# Effects of Microwave-Assisted *Opuntia humifusa* Extract in Inhibiting the Impacts of Particulate Matter on Human Keratinocyte Skin Cell

**DOI:** 10.3390/antiox9040271

**Published:** 2020-03-25

**Authors:** Ju-Young Moon, Le Thi Nhu Ngoc, Minhe Chae, Vinh Van Tran, Young-Chul Lee

**Affiliations:** 1Department of BioNano Technology, Gachon University, 1342 Seongnam-Daero, Sujeong-Gu, Seongnam-Si, Gyeonggi-do 13120, Korea; bora7033@naver.com (J.-Y.M.); nhungocle92@gmail.com (L.T.N.N.); 2Biocell Korea Co., Ltd., 1-2FJanghan B/D, 54 Bongeunsa-ro 30-gil, Gangnam-gu, Seoul 04631, Korea; christinechae7@gmail.com

**Keywords:** *Opuntia humifusa*, microwave-assisted extract, particulate matter, anti-pollutant, plant extract

## Abstract

Particulate matter (PM) is one of the most important factors causing serious skin diseases, due to its generation of reactive oxygen species (ROS) over the course of long-term exposure. As a source of natural antioxidants, *Opuntia humifusa* (*O. humifusa*) is a potential candidate for the design of advanced formulations to prevent PM’s harmful effects. Unfortunately, its high viscosity does not allow it to be utilized in these formulations. In this present study, a new approach to the extract of *O. humifusa* using high-power microwave treatment, namely microwave-assisted *O. humifusa* extract (MA-OHE), was investigated. The results indicated that MA-OHE not only is a reasonable viscosity extract, but also enhances *O. humifusa*’s antioxidant properties. Additionally, this study also found that MA-OHE exhibited outstanding antioxidant and anti-inflammatory activities in eliminating PM’s effects, due to suppression of AhR degradation, ROS production, and COX-2 and MMP-9 expression in HaCaT keratinocytes. It is believed that MA-OHE is a potential cosmeceutical ingredient that could be utilized to prevent PM-induced skin oxidative stress and inflammation.

## 1. Introduction

Recently, particulate matter (PM), which comprises various organic and inorganic particles of micrometer size, was estimated to account for the highest proportion of air pollution [1]. PM commonly originates from volcanoes, forest fires, grassland fires, and human activities (e.g., industrial processes, power plant operation, and vehicle emissions) [1]. Based on the diameter that can cause harmful effects to humans, PM is primarily classified into two group: PM_10_ (≤10 µm) and PM_2.5_ (≤2.5 µm) [1,2,3]. At such small sizes, they can easily penetrate and deposit in the respiratory tract, including the bronchi, trachea, and deep lungs [4,5], causing many health problems such as cardiovascular diseases (CVD), respiratory health effects, diabetes, and especially, skin diseases [5,6]. According to a WHO report (2013), PM pollution is associated with about 4.2 million premature deaths each year, including from lung cancer (16%), ischemic heart and stroke (17%), chronic obstructive pulmonary disease (COPD) (25%), and respiratory infection (26%), and is ranked the 14th risk of death worldwide [7]. In particular, PM has been revealed to be the major factor in the onset of serious skin diseases, including atopic dermatitis, acne, pigmentation spots, psoriasis, skin aging, and wrinkles [8,9]. Its underlying, long-term biological mechanism entails the generation of reactive oxygen species (ROS), overexpression of pro-inflammatory mediators, and eventually, disruption of skin integrity [9].

The skin, the largest organ of the human body, acts as the largest interface between the body and the external environment. Therefore, it is necessary to prevent skin diseases due to external factors, especially PM pollution. Recently, a number of biomedical and cosmetic products have been applied to protect the skin, as well as improve aesthetic appearance. Their biomaterials, which originated in or are synthesized from plant extracts, offer several outstanding physical and chemical properties such as high antioxidant activity and high biocompatibility that incur few side effects [10]. Therefore, this study aimed to develop a novel material from plant extracts for further cosmetic applications.

Among biomedical plants, *O. humifusa*, known as Korean Cheonnyuncho, is rich in nutrients (i.e., betalains, vitamins, minerals, phenolic compounds, flavonoid compounds, and polysaccharides) [11,12,13,14]. Therein, phenolic acids include vanillic acid, p-coumaric acid, ferulic acid, syringic acid, 4-hydroxybenzoic acid, protocatechuic acid, salicylic acid, caffeic acid, gallic acid, and sinapic acid, while flavonoid compounds include rutin, narcissi, kaempferol, and quercetin [15]. Based on *O. humifusa*’s outstanding features (e.g., high moisturization, high antioxidant activity, anti-inflammation, and anti-diabetes), its extract has been widely applied for topical-cosmetic, biomedical, and food-additive purposes [11,12,13]. For example, Yoon et al. (2009) reported that *O. humifusa* had been successfully applied to treat various diseases, including arteriosclerosis, gastritis, diabetes mellitus, and hyperglycemia [16]. Ha et al. (2016) pointed out that *O. humifusa* extracts effectively inhibited the transcription factor associated with microphthalmia, reduced-matrix metalloproteinase-1 (MMP-1), and phosphorylation of JNK. It is suggested, in fact, that they are valuable as ingredients in various functional cosmetics, imparting whitening and wrinkle-improvement effects [13]. In another study, Park et al. (2017) investigated the skincare application potential of *O. humifusa* for facilitating inhibition of water loss and erythema formation by regulating UVB-induced hyaluronic acid production [11].

In particular, *O. humifusa* is known to contain soluble dietary fiber and an abundance of calcium and methoxyl pectin, the constituents of which endow it with strong physical properties (e.g., water-retention capacity, swelling, high viscosity) that could be potential features to use as food additives [17,18]. However, its high viscosity seems to be a drawback respecting its cosmetic applications, a fact that demands the development of a viscosity-reducing extraction method. Fortunately, it has been reported that microwave treatment can alter the physicochemical properties of gelatinized plant extracts by viscosity reduction, as well as alteration of the crystalline structure, morphology, and molecular weight distribution [19,20]. In fact, microwave energy can affect dipole rotation, subsequently disrupting cell walls and breaking down longer dextran chains, thereby releasing the expected compounds and reducing the viscosity of the extract solution [20,21]. In fact, the microwave-assisted extraction method has been widely utilized for its rapidity and high efficiency [21]. The present study, therefore, aimed to apply microwave assistance in order to reduce the viscosity and enhance the effectiveness of *O. humifusa* extract. Additionally, it undertook investigating the potential of *O. humifusa* extract as an anti-pollution ingredient for cosmetic applications protecting the skin from PM pollution.

## 2. Materials and Methods

### 2.1. Reagents

*O. humifusa* (Korean Cheonnyuncho) grown in Asan, Korea, was harvested and washed with water (later removed with paper towels) and stored in a refrigerator. Dimethyl sulfoxide (DMSO, 78.13 g/mol), methanol, Folin–Denis’s reagent, quercetin, gallic acid, ethylenediaminetetraacetic acid (EDTA), the Annexin V-FITC apoptosis detection kit, 2,2-diphenyl-1-picrylhydrazyl (DPPH, 394.32 g/mol), 2′,7′-dichlorofluorescin diacetate (DCFH-DA, 487.29 g/mol), l-glutamine 200 mM (100×), penicillin/streptomycin (Pen/Strep, 100×), Cell Proliferation Kit I (MTT), trizma base, triton X-100, and sodium chloride (NaCl, 58.44 g/mol) were purchased from Sigma-Aldrich (St. Louis, MO, USA). Dulbecco’s Modified Eagle Medium (DMEM) was supplied from Welgene (Gyeingsangbuk-do, Korea). Fetal bovine serum (FBS) and trypsin/EDTA were obtained from Gibco^®^ via Life Technologies (Rockville, MD, USA). Dulbecco’s phosphate-buffered saline (DPBS), RIPA-buffer, Tris-buffer, Tween-20 TBST, primary anti-body COX-2, and chemiluminescence agent were supplied by Thermo Fisher Scientific (Waltham, MA, USA). Aluminum chloride hexahydrate (AlCl_3_·6H_2_O; 241.43 g/mol), sodium hydroxide (NaOH; 39.997 g/mol), and sodium carbonate (Na_2_CO_3_; 105.98 g/mol) were purchased from Daejung Co., Ltd. (Busan, Korea). Hydrogen peroxide (35%) was obtained from Junsei Chemical Co., Ltd. (Tokyo, Japan). GelRed^®^ nucleic acid gel stain was provided by Biotium, Inc. (Fremont, CA, USA). Primary antibody p-ERK 1/2 (12D4), beta-actin, Ah receptor (A-3), and CYP1A1 (A-9) were supplied by Santa Cruz Biotechnology, Inc. (Dallas, TX, USA). Primary anti-body MMP-9 was obtained from Abcam (Cambridge, U.K.). Milli-Q distilled water was utilized throughout the experiments (conductivity < 18.2 MΩ cm^2^; Milli-Q Millipore filter system, Millipore Co., MA, USA).

According to our previous study [3], PM samples were collected from subway stations in Seoul, Korea. The samples were collected on Φ 47 mm quartz filters (QMA filter, Φ 47 mm, Pall Corporation, NY, USA) utilizing Ze-flour filters (polytetrafluoroethylene membrane filter, Φ 47 mm, Pall Corp.) and a mini-volume air sampler (TAS, 5 L per min, Airmetrics, Eugene, OR, USA) using a low-volume air sampler (PMS-104, 16.7 L per min, APM, Bucheon, Korea). PM samples were dispersed in DI water for 30 min by using a sonication bath, and the PM dispersions were used within 1 h to avoid the degradation of PM components.

### 2.2. Microwave-Assisted O. humifusa Extract

For the preparation of raw *O. humifusa*, fresh and healthy leaves of *O. humifusa* were thoroughly washed with water, air dried, and chopped into small pieces. The chopped leaves (20 g) were then ground into a homogenous mixture in a blender. The resulting product had a very high viscosity, which was directly used for viscosity measurement and production of microwave-assisted *O. humifusa* extract (MA-OHE) extract. For further experiments and assays, *O. humifusa* powder, produced by a freeze-drying process, was used and diluted with DI water in various concentrations.

The domestic microwave oven (Magic MMO-20M7) utilized in this study possessed a total capacity of 800 W. The MA-OHE process was conducted according to the method of Alvand et al. (2019) with some modifications (Figure 1) [22]. The raw *O. humifusa* with very high viscosity was irradiated with microwaves (800 W power) for 10 min. After irradiation, the solid sample was allowed to cool at room temperature and ground down. DI water (15 mL) was added to dissolve active agents, and insoluble impurities were removed by the process of centrifugation. The suspension was filtered quickly through a 0.22 µm centrifuge tube filter to obtain a pure extract solution. Finally, the pure extract powder of MA-OHE was obtained by freeze-drying of the extract solution. Prior to its analysis, MA-OHE powder was dissolved in DI water with various concentrations.

The extraction yield Y (%) of MA-OHE was calculated as follows:(1)$$Y\left(\%\right)=\frac{W_{1}}{W_{2}}\times 100$$
where *W*_1_ is the weight (g) of pure MA-OHE powder, while *W*_2_ is the weight (g) of the initial *O. humifusa* powder. The calculated result indicated that the extraction yield was about 33.5%.

### 2.3. Viscosity Measurement of MA-OHE

Dynamic viscosity measurement was carried out at 25 °C and atmospheric pressure using a Haake viscometer (VT550, Karlsruhe, Germany) [17]. The applied shear rate was 100 s^−1^ at 25 °C for 60 s. The MA-OHE extract powder was gradually dissolved by DI water to make suspensions under gentle mechanical stirring at 25 °C for 2 h. Before measurements were taken, these suspensions were allowed to equilibrate at 25 °C for 4 h. Glycerol (10%) and a suspension of raw *O. humifusa* were used for comparison. Each measurement was replicated three times, and the uncertainties of the viscosity values were within 0.01 mPa s.

### 2.4. Antioxidant Content Analysis of MA-OHE

The total polyphenol content was measured using Folin–Denis reagent. Briefly, 100 µL of MA-OHE were added to a 15 mL conical tube and then topped up to 1 mL by DI water. Then, 100 µL of Folin–Denis reagent were added to this solution, followed by reaction at room temperature. After 3 min of reaction, 200 µL of 10% Na_2_CO_3_ were subsequently added, and then the mixture reaction solution was topped up to 2 mL by DI water. The final mixture was incubated for 1 h in darkness. Finally, the supernatant was measured for absorbance at 760 nm using UV-Vis spectroscopy. The total polyphenol content was calculated using standard calibration curves of gallic acid (GA).

The total flavonoid content was measured using methods described in Woisky and Salatino [23] with a slight modification. Methanolic solutions of quercetin (5–50 µg/mL) were used as references. Briefly, 0.6 mL of each reference solution and MA-OHE extract (50 mg/mL) were added to 0.6 mL of 2% AlCl_3_. The mixtures were allowed to react at room temperature for 1 h. After 1 h, the absorbance was measured at 420 nm using UV-Vis spectroscopy. The total flavonoid content was calculated based on a standard quercetin calibration curve.

### 2.5. Antioxidant Activity Analysis

#### 2.5.1. Preparation of Samples

The antioxidant activity of the pure MA-OHE powder was tested by DPPH radical and hydroxyl radical assays. The pure MA-OHE powder was dissolved and diluted in DI water to specific concentrations under gentle shaking. In comparison, the raw *O. humifusa* samples were simultaneously tested with antioxidant activity assays. The raw *O. humifusa* was also dissolved in DI water with high-speed stirring. However, corresponding to each concentration of the pure MA-OHE powder, the raw *O. humifusa*’s concentration was tripled. This preparation method was also used for cell viability assays.

#### 2.5.2. DPPH Radical Scavenging Activity

DPPH radical scavenging ability was measured using methods described by Brand-William et al. (1995) [24]. Briefly, the pure MA-OHE powder was dissolved and diluted to final concentrations of 100, 200, 500, 1000, 3000, and 5000 µg/mL. A total of 0.1 mL of each extract solution was added to 0.9 mL of DPPH free radical (0.1 mM) in methanol. After 30 min, the absorbance was measured at 517 nm. Finally, the antioxidant activity was calculated by Formula (2).
(2)$$\mathrm{DPPH}•\;\mathrm{scavenging}\;\mathrm{efficiency}\;(\%)=\frac{A_{0}-A}{A_{0}}\times 100\%$$
where *A_0_* and *A* are the absorbance of the DPPH• radical solution at 517 nm in the presence of the control sample and the extract samples, respectively.

#### 2.5.3. Hydroxyl Radical Scavenging Activity

OH• radical scavenging activity was measured by a modified version of the salicylic acid method [25,26]. Briefly, OH• radical solution was obtained by mixing three solutions including salicylic acid (9 mmol/L), FeSO_4_ (9 mmol/L), and hydrogen peroxide (9 mmol/L), and then, 0.3 mL of each extract sample (in different concentrations) were added to 0.9 mL of OH• radical solution. The reaction mixture was incubated for 30 min at 37 °C, and the absorbance was determined at 510 nm. The hydroxyl radical scavenging activity (%) of the sample was determined by Formula (3):(3)$$\mathrm{OH}•\;\mathrm{radical}\;\mathrm{scavenging}\;\mathrm{efficiency}\;\left(\%\right)=\frac{A_{0}-A}{A_{0}}\times 100\%$$
where *A*_0_ and *A* are the absorbance of the OH• radical solution at 510 nm in the presence of the control samples and extract samples, respectively.

### 2.6. Cell Culture

Human keratinocyte cells (HaCaT) were obtained from the Korean Cell Line Bank (Seoul, Korea). According to the literature [27], the HaCaT cells were cultured in growth DMEM medium (1% Pen/Strep, 1% l-glutamine, and 10% FBS). These cells were sub-cultured every other day when the cell density reached 70–90% confluence. Firstly, after removing the supernatant, these cells were twice washed with DPBS buffer, and then, trypsin/EDTA was added to detach them from the bottom of the flask. Next, the cells were mixed with new growth DMEM medium, collected in conical tubes, and then, centrifuged for 3 min at 1200 rpm. Finally, the cell destiny was adjusted to 10^5^–10^6^ cells/mL, and these cells were incubated consecutively in T-25 culture flasks (37 °C and 5% CO_2_).

### 2.7. Cell Viability Assay

In this study, an MTT assay was performed to determine the cell viability effect of MA-OHE, raw *O. humifusa,* and PM. Briefly, the cells (1 × 10^5^ cells/mL) were seeded in a transparent 96 well plate (Falcon, Franklin, NJ, USA) and incubated (37 °C and 5% CO_2_) for 24 h. After 24 h of incubation, the growth DMEM medium was removed and replaced by 100 µL of MA-OHE, raw *O. humifusa*, and PM at different concentrations (0, 10, 20, 50, 100, 200, and 300 µg/mL). For evaluating the effects of MA-OHE and raw *O. humifusa* on the PM-induced cell survival ratio, cells were treated with 100 µg/mL for 2 h, and then, different concentrations of MA-OHE and raw *O. humifusa* were inserted.

After 24 h, 10 µL of MTT reagent (0.5 mg/mL) were added to each well, and the plate was incubated for 4 h (37 °C and 5% CO_2_). Then, 100 µL of solubilization solution were added to each well, and the plate was incubated overnight (37 °C and 5% CO_2_). Finally, the absorbance was measured at a wavelength of 590 nm using a microplate reader (Multi-label plate reader, PerkinElmer, Boston, MA, USA).

### 2.8. Evaluation of Intracellular Reactive Oxygen Species

The generation of intracellular ROS was measured by the DCFH-DA assay. Briefly, HaCaT cells (2.5 × 10^5^ cells/mL) seeded in a black 96 well plate (Thermo Scientific™, Waltham, MA, USA) were pretreated by MA-OHE and raw *O. humifusa* at a concentration of 0–200 µg/mL for 2 h; then exposed to PM (100 µg/mL) [3]. After 1 and 2 days of incubation, 10 µM DCFH-DA dissolved in DMSO were stained for 1 h in a darkroom and then twice washed with PBS buffer. In the literature, DCFH-DA penetrates into the internal cells and then is hydrolyzed into DCFH by esters. Immediately, this was recorded by fluorescence measurement at 485/530 nm. Dipyridamole was used as a control with the relative ROS level of 100%. The ROS generation was calculated as Formula (4)
(4)$$\%\mathrm{ROS}\;\mathrm{gerneration}=\frac{F}{F_{o}}\times 100$$
where *F* is the fluorescence intensity of cells pretreated with MA-OHE and raw *O. humifusa*; *F_o_* is the fluorescence intensity of dipyridamole.

### 2.9. Apoptosis Detection

In this study, the Annexin V-FITC/PI staining kit was used to determine the externalization of phosphatidylserine during early apoptosis induced by MA-OHE and PM exposure [28]. Briefly, after 24 h of incubation in a 6 well plate, HaCaT cells (1 × 10^6^ cells/mL) were treated with 100 µg/mL of PM; then, MA-OHE and raw *O. humifusa* were added and incubated for 24 h. After these treatment periods, the cells were collected and washed two times with PBS buffer, after which they were suspended in a binding buffer and stained with single Annexin V-FITC, single PI, and a mixture of Annexin V-FITC and PI in a FACS holder before detection of apoptosis by the CellQuest program of FACS machine (BD FACS Calibur, Becton, Dickinson and Company, San Jose, CA, USA).

### 2.10. Genotoxicity Determination

According to the standard protocol [29], a comet assay was conducted to assess the genotoxicity of MA-OHE and PM to HaCaT cells. The cells were cultivated in a 6 well plate for 24 h. After that, cells were treated with PM (100 µL) for 24 h, and then, MA-OHE and raw *O. humifusa* were added. For the positive control, HaCaT cells were treated with H_2_O_2_ (10 µM) for 10 min at 4 °C. After the treatment period, the cells were twice washed with DPBS buffer, stripped off by trypsin/EDTA, and then suspended in DPBS buffer (with Mg^2+^ and Ca^2+^) at a density of 5 × 10^4^ cells/mL. Next, a mixture of 20 µL cell suspension with 60 µL of 0.5% low-melting agarose (LMA; Lonza, Switzerland) was layered onto microscope slides pre-coated with 0.1% normal melting agarose (NMA; Sigma-Aldrich, St. Louis, MO, USA). The slides were covered with coverslips and kept on ice for 15 min to solidify the agarose. After solidification, the coverslips were removed, and 80 mL of molten 0.75% LMA prepared in DPBS were spread onto the slides. The slides were again covered with coverslips and kept on ice for 15 min. Then, the coverslips were removed, and the slides were immersed in cold, freshly made lysing solution (2.5 M NaCl, 100 mM EDTA, 10 mM Trizma base, pH 10, and containing 1% Triton X-100) for 24 h at 4 °C in a dark chamber. All of the slides were placed for 20 min in a horizontal gel electrophoresis tank filled with cold electrophoresis buffer (0.3 M NaOH and 1 mM EDTA, 4 °C) to allow for DNA unwinding. Then, electrophoresis was performed in the same buffer for 20 min at 25 V (1 V/cm) and 300 mA at 4 °C. After that, the slides were neutralized with three washes for 5 min in fresh chilled Tris buffer (400 mM, pH 7.5). Prior to the analysis, the slides were stained with GelRed dye for 30 min in darkness. Comet appearances were analyzed using a fluorescence microscope (Nikon ECLIPSE E400, Tokyo, Japan) at 400× magnification. Then, 50 randomly selected cells (three independent experiments) were analyzed per treatment. The percentage of DNA in the tail and olive moment value were used to evaluate the extent of DNA damage by Cometscore software (Tritek Corp., Samerduck, VA, USA).

### 2.11. Determination the Effect of MA-OHE on Reducing the PM-Induced HaCaT Keratinocyte Injury

The PM-induced HaCaT cells model was built as described in Huang et al. [30]. Briefly, PM was suspended in PBS and then sonicated for 30 min in a sonication bath. HaCaT cells (1 × 10^6^ cells/mL in a 6 well plate) were treated with MA-OHE (50, 100, 200 µg/mL) for 2 h. After that, 300 µg/mL of PM were added to each well for 24 h. Experiments were undertaken within 1 h of PM preparation to prevent variations in PM composition. Then, the cells were twice washed with PBS, and the total proteins were lysed with RIPA buffer in ice (pH 7.4; protease inhibitor, 1 mM phenylmethylsulfonyl fluoride, and 1 mM sodium orthovanadate). For the purposes of the Western blotting assay, sodium dodecyl sulfate-polyacrylamide gel electrophoresis was performed using 10% running gel, and then, the proteins were transferred onto poly(vinylidene fluoride) membranes. The membranes were incubated overnight at 4 °C with primary antibodies including MMP-9 (1:1000), COX-2 (1:1000), AhR (1:1000), CYP1A1 receptor (1:1000), and β-actin (1:2000), washed with Tris-buffered saline with Tween-20 (TBST), incubated with anti-mouse, anti-rabbit, and anti-goat HRP-conjugated secondary antibodies for 30 min at room temperature, and then, washed three times with TBST again. The immunoreactive bands of each sample were reacted with enhanced chemiluminescence reagents and visualized using the ChemiDoc MP imaging system (Bio-Rad Laboratories, Inc., Berkeley, CA, USA).

### 2.12. Statistical Analysis

All of the data were analyzed by Microsoft Excel 2016 software (Microsoft Office, Microsoft Corporation, Redmond, WA, USA) and expressed as the mean ± SD. All graphs were drawn using Origin plotted software (Version 8.5, OriginLab Corporation, Northampton, MA, USA). Significant differences of all data were analyzed using one-way and two-way ANOVA for single treatment comparisons (GraphPad Prism software, San Diego, CA, USA).

## 3. Results

### 3.1. Reduced Viscosity Effect of O. humifusa Extract by Microwave Treatment

The rheological properties of plant extracts are of importance when designing new formulations. Among them, viscosity usually is used to determine their suitable formulations, because viscosity not only affects some crucial characteristics (i.e., skin feel and spreadability), but might also impact the skin penetration of incorporated active agents [31]. A reasonable viscosity of plant extracts is beneficial for the development of cosmetic formulations for the skin. Generally, glycerin is widely used in the food industry and in pharmaceutical (i.e., drug) and cosmetic formulations. Although glycerin can be used at concentrations up to 99.4% in some cleaning products, the commonly used concentrations of glycerin are below 40%. Therefore, in the present study, a viscosity comparison of glycerin 10%, 20%, and 40% with raw *O. humifusa* and MA-OHE extract was performed. The measured viscosity values are presented in Figure 2. The results indicated that the viscosity considerably decreased from 128 mPa·s (raw *O. humifusa)* to 2.35 mPa·s (MA-OHE) under microwave treatment. This reduction may have been due to reduced attraction forces between the molecules caused by the high-power microwave radiation. In addition, the viscosity of the MA-OHE extract was found to be similar to that of glycerin solution (20%). These results indicated that the microwave-assisted process was a suitable method for the reduction of *O. humifusa*’s high viscosity and that as-prepared MA-OHE extract could be used in cosmeceutical and pharmaceutical formulations requiring a low viscosity, such as essence, sprays, aerosols, shampoos, etc.

### 3.2. Antioxidant Activities of MA-OHE

The antioxidant activities of MA-OHE and raw *O. humifusa* were determined by scavenging efficiency against DPPH• and OH• radicals in a comparison with the antioxidant standards (ascorbic acid) (Figure 3). The values were the mean ± SD (*n* = 3). Overall, the antioxidant activities of MA-OHE and raw *O. humifusa* showed a concentration-dependent relationship. The MA-OHE exhibited slightly lower DPPH• and OH• radical scavenging activities than did ascorbic acid, with the former already reaching maximal inhibition of the DPPH• and OH• radicals at concentrations above 1000 and 1600 µg/mL, respectively. Meanwhile, the DPPH• and OH• radical scavenging activities of the raw *O. humifusa* were very low, compared with MA-OHE, and they only showed a slight increase with the increase of the concentration. Generally, the inhibitory activities of MA-OHE extract were rapidly increased at concentrations below 400 µg/mL, after which the reaction slowed down significantly until the highest concentration. In another study, this one by Cha et al. (2013) [32], a similar result was reported for the DPPH• radical scavenging activity of *O. humifusa* fruit-based extracts using solvent extract methods with acetone, ethanol, and methanol. The MA-OHE reached EC_50_ values for both DPPH• and OH• radical assays at 300 and 600 µg/mL. These results from the two antioxidant assays suggested that MA-OHE extract was a potent antioxidant and that microwave-assisted extract could be an effective method for the extract of *O. humifusa* leaves. In addition, the MA-OHE was detected to contain large amounts of phenolic and flavonoid compounds, respectively, about 33.1 and 4.06 mg/g, significantly higher than those amounts extracted by other extraction methods (~50.6–149.8 µg/g) in previous studies [32,33]. In comparison with the conventional methods (hot water extraction and solvent extraction), it can be said that the microwave-assisted extraction method offered potential efficacy in reducing viscosity, as well as enhancing antioxidant contents.

### 3.3. Effect of MA-OHE on Reducing the Toxicity of PM to Keratinocyte Skin Cells

#### 3.3.1. Effects of MA-OHE on PM-Affected Cell Viability

First of all, the biocompatibility of MA-OHE and raw *O. humifusa*, as well as PM was assessed by the MTT assay (Figure 4A). It was shown that MA-OHE resulted in negligible toxicity in both high- and low-dose treatments (>8 % cell survival), while raw *O. humifusa* showed a slightly higher toxicity on HaCaT cells after 24 h treatments. It can be said that the microwave-assisted extract offered greater biocompatibility than did the raw *O. humifusa*; therefore, MA-OHE instead of raw *O. humifusa* was subsequently assessed by other biological assays for further anti-pollutant applications. Additionally, the present study also investigated the cytotoxicity of PM on keratinocyte cells. It was found that PM had an obvious impact on the proliferation of HaCaT cells, which were significantly reduced to less than 80% at high concentrations (>100 µg/mL) in the 24 treatments.

In the present study, the MTT assay was also used to assess the effectiveness of MA-OHE and raw *O. humifusa* in reducing the impact of PM on HaCaT cells. Briefly, HaCaT cells were pretreated with MA-OHE (10, 20, 40, 60, 80, and 100 µg/mL) for 2 h and then exposed to PM (100 µg/mL) for 24 h. According to the results, the cells that were non-pretreated and exposed by PM showed a survival ratio of 85%, and both the MA-OHE and raw *O. humifusa* pretreatments showed a positive effect in inhibiting the toxicity of PM to cells (Figure 4B). It was found that the pretreated cells by MA-OHE exhibited a higher survivability than the raw *O. humifusa*. Specifically, the survival of HaCaT cells increased with increasing MA-OHE pretreatment concentrations from 10 to 100 µg/mL. At the highest concentrations (100 µg/mL), MA-OHE showed the best potential to protect cells from the harmful effects of PM (≥98% of cell survival). On the other hand, the pretreated cells with the raw *O. humifusa* only exhibited a small increase (2.5%) in the survival ratio as concentration of *O. humifusa* increased from 10 to 100 µg/mL.

#### 3.3.2. Effects of MA-OHE in Reducing PM-Induced ROS and Apoptosis

The effects of MA-OHE and raw *O. humifusa* in reducing PM-induced ROS radicals resulting in DNA damage were determined using a dose-dependent DCFH-DA assay (Figure 5A). According to the results, PM (100 µg/mL) generated a huge number of ROS radicals during short-term exposure, reaching 138.5% in DCFH-DA fluorescence versus control samples, a fact that proved that it was at high risk of causing apoptotic and necrotic cell death of HaCaT cells [3]. MA-OHE exhibited an excellent ROS scavenging capacity (≥40%) at a concentration of 200 µg/mL. Conversely, raw *O. humifusa* showed a low ROS scavenging capacity of 15% a at concentration of 200 µg/mL.

Furthermore, the mode of cell death on HaCaT cells caused by PM and the effects of MA-OHE and raw *O. humifusa* in reducing the cell death were studied through flow-cytometry using Annexin V-FITC/PI staining. Under such staining, early-/late-apoptosis cells, as introduced from the inside to the outer leaflet of the plasma membrane during the apoptosis stages, could easily be detected [34]. Meanwhile, the PI could only stain necrotic cells, because it depended on membrane permeability to enter the cells [34]. In this study, the percentages of the apoptotic and necrotic populations of treated and untreated HaCaT cells were determined based on the flow-cytometric outcomes. As shown in Figure 5B, MA-OHE pretreatment induced the lowest apoptosis incidence (8.6%) during 24 h of treatment, relative to the control cells (~8.7%). On the other hand, non-pretreatment and PM exposure (100 µg/mL) led to an increase in the incidence of the apoptosis and necrosis of HaCaT cells, which was about 1.5-fold higher than for MA-OHE-pretreated cells in both apoptosis and necrosis. Meanwhile, the data of raw *O. humifusa* pretreatment also indicated a slight decrease in the proportion of apoptosis and necrosis, compared with the non-pretreatment sample, 19% and 27% respectively; but it was higher than that of MA-OHE pretreatment. It could be explained by the fact that MA-OHE contained a larger number of antioxidants than raw *O. humifusa*, which could protect cells from the destruction of external factors, as well as ROS radicals on the cell surface. By contrast, PM mainly consisted of heavy metals and organic compounds that could produce a huge number of ROS radicals, leading to DNA damage and, thereby, increasing the extent of apoptosis, as well as the number of cell death events [3].

#### 3.3.3. Effects of MA-OHE in Reducing PM Genotoxicity

The comet assay was used to assess the effects of MA-OHE and raw *O. humifusa* in reducing PM genotoxicity in single HaCaT cells. Its principle is to use gel electrophoresis to separate damaged DNA, which is then visualized by DNA binding fluorescent probes. The image analysis determines both intact DNA (comet head) and fractured DNA (comet tail). It was revealed that the greater the damage, the greater the proportion of DNA in the tail and tail olive moments. Before conducting the genotoxicity analysis of the selected doses of PM using the comet assay, the toxicity of the selected doses was evaluated by the MTT assay. The results indicated that both treatments at the selected concentrations (100 µg/mL) always produced higher (70%) cell survival.

The effects of MA-OHE and raw *O. humifusa* in reducing PM genotoxicity of HaCaT cells in terms of tail DNA length and tail olive moment are plotted in Figure 6A. In the MA-OHE-containing systems, the tail length of single HaCaT cells (~10.26%) was slightly higher than that of the negative control (4.63%), and this proportion was obviously lower than for the positive control (~52.31%) and raw *O. humifusa* pre-treatment (16.74%) (Figure 6B,C). In contrast, PM induced significant genotoxicity to HaCaT cells, the effect of which was shown in terms of the percentages of both DNA tail and tail olive moment relative to the MA-OHE treatment and control groups.

### 3.4. Potential of MA-OHE for Inhibition of Pro-Inflammatory Protein Expression Induced by PM Exposure

Firstly, to determine whether PM treatment led to AhR activation, the expression of the biomarker of AhR (cytochrome P450–CYP1A1) was investigated (Figure 7A,B). The data obtained showed that AhR was degraded, resulting in over-stimulation of CYP1A1 protein in HaCaT cells under the treatment of PM (100–300 µg/mL). On the other hand, pretreatment of HaCaT cells with MA-OHE prior to PM exposure showed a potential to protect against AhR degradation and, consequently, to downregulate the expression of CYP1A1 protein. In particular, the 200 µg/mL MA-OHE pretreatment completely prevented keratinocyte damage and reduced AhR degradation 3.4-fold relative to the un-pretreated cells.

It is well known that PM also causes significant inflammation in human skin, which is mediated by overexpression of pro-inflammatory proteins (MMP-9 and COX-2) and MAPK signaling (ERK1/2). Therefore, the present study assessed the effects of MA-OHE in reducing the expression of these proteins. According to the results, PM obviously behaved as an inflammation inducer to increase the pro-inflammatory protein level after 24 h of exposure. However, fortunately, the levels of MMP-9 and COX-2 were significantly decreased by approximately 3.87- and 7.2-fold by pretreatment with MA-OHE, respectively, as compared with the non-pretreatment group (PM 500 µg/mL) (Figure 7C,D). Moreover, by MA-OHE pre-treatment for 1 h before exposure to PM (200 µg/mL), the level of ERK1/2 also was slightly downregulated (Figure 7E).

## 4. Discussion

Antioxidant activity is one of the most important features to evaluate the applicability of plant extracts [35,36]. Although *O. humifusa* contains an abundance of antioxidant components that are excellent candidates for cosmetic applications, it has not been widely applied as a cosmetic ingredient due to its high viscosity [17]. Therefore, this study aimed to develop a simple and effective method to reduce the viscosity of *O. humifusa* by means of microwave assistance. According to the results, the MA-OHE presented not only low viscosity, but also enhanced antioxidant capacity (phenolic and flavonoid compounds) relative to raw *O. humifusa*. As compared with the conventional methods, MA-OHE exhibited several remarkable advantages regarding the yield, viscosity, and antioxidant activity [11,12,17,21]. For illustration, MA-OHE showed a significantly higher yield (33.5%), compared with extracts prepared by conventional methods such as solvent extraction and the hot water method (10.5–23.1%) [11,17]. The data also indicated that MA-OHE possessed a higher antioxidant activity, nearly 98% of DPPH radical scavenging efficiency, while the maximum percentage of DPPH radical scavenging efficiency reported in other studies was only 80% [32]. While MA-OHE showed similar effects to those of other advanced technologies (e.g., ultrasound-assisted extraction (UAE)) in terms of yield (UAE: 31.7%) [12], MA-OHE possessed a lower viscosity as compared with the UAE method. This was due to the fact that the acoustic energy (mechanical energy) used in the UAE process was not absorbed by molecules, but was only transmitted throughout the medium [21], whereas the microwave method used high energy to heat molecules by a dual mechanism of ionic conduction and dipole rotation, which reduced the attractive forces between the molecules.

In addition, the cytotoxicity results indicated that MA-OHE possessed greater biocompatibility than did the raw *O. humifusa*, slightly reducing the cell survival of HaCaT cells from 87.47 ± 1.60 and 80.88 ± 4.84% after 24 h treatment at 100 µg/mL. A possible explanation was that under high-temperature and high-energy treatment, the raw *O. humifusa* was transformed into carbon dots (CDs) [22] and significantly reduced the viscosity. It was supposed that it was the viscosity reduction that allowed MA-OHE to be able to release antioxidant compounds, which resulted in a higher antioxidant activity for MA-OHE, compared with raw *O. humifusa.* Moreover, it was demonstrated that the plant extract-based CDs prepared by the microwave-assisted process had low toxicity [22]. Overall, the microwave-assisted extraction method helped to maintain nutrition, improve texture and appearance, and reduce viscosity relative to the conventional extraction methods. All of this was potential evidence for MA-OHE’s superior potentiality vis-à-vis raw *O. humifusa* for cosmetic applications. Regarding the antioxidant activity that was attributed to huge amounts of flavonoid, phenolic, and CD fractions, the MA-OHE exhibited great biocompatibility, including non-toxicity and induction of negligible apoptosis and genotoxicity. In particular, the antioxidants in MA-OHE presented a preventive role in the scavenging of ROS free radicals; indeed, those antioxidants could minimize apoptosis and the genotoxicity of HaCaT cells over the course of long-term treatment [10,35]. In addition, flavonoid and phenolic compounds are considered to be effective anti-inflammatory factors as well [37,38,39]. It has been reported that these bioactive extracts exert their biological properties by two major signaling pathways, including mitogen-activated protein kinases (MAPKs) and phosphorylation transcript factor (NF-κB), which play a critical role in the production of various pro-inflammatory mediators [37,39].

On the other hand, the impacts of PM in terms of skin damage, especially its underlying molecular biological mechanisms, have been clearly investigated in several studies. There is no denying the fact that PM pollution increases the incidence of skin diseases, including inflammation, androgenic alopecia, extrinsic aging, and skin cancer [8,9]. Furthermore, it has been reported that PM is a major constituent of transition metal fractions (Fe, Cu, Zr, Ba, and V) and organic compounds (quinones and polycyclic aromatic hydrocarbon), which are primarily responsible for the negative effects of PM [3,40]. Certainly, transition metals play an important role in the production of ROS radicals on the cell surface through Fenton reactions [3,41]. These organic fractions also can undergo redox-cycling, producing hydroxyl radicals and promoting the expression of cytochrome enzymes that lead to increased ROS production [41,42]. In fact, the primary underlying mechanism of PM-induced skin diseases has been attributed to the generation of ROS (hydroxyl and superoxide radicals), activation of AhR, and production of inflammatory mediators [43]. Therein, the activation of AhR caused by PM is regulated by upregulation of cytochrome P450 (CYP1B1, CYP1A1, and CYP1A2) [43]. Besides, ROS generation is mediated through activation of MAPKs signals (ERK1/2, p38, and JNK), phosphorylation transcription factors (NF-κB and activator protein-1), and upregulation of the expression of pro-inflammation mRNA (MMPs, COX-2, and collagen types I and III) [44,45]. In addition, the overexpression of inflammatory cytokines (TNF-α, IFN-γ, IL-6, IL-8, IL-1α, and IL-1β) also plays a key role in PM-inducing oxidative stress that results in lipid peroxidation and DNA damage [46,47,48]. Another major mechanism entails endoplasmic reticulum stress, autophagy, and mitochondrial swelling caused by PM exposure, which increases the expression of pro-apoptotic markers (caspase-3/-9 and cytochrome c) and leads, thereby, to apoptosis that manifests as skin damage [49,50].

Indeed, PM had a significant effect on the skin, reducing the cell viability of HaCaT cells (<80%), generating a large number of ROS radicals (~1.47-fold higher versus control group), and remarkably promoting apoptosis and genotoxicity incidences. Moreover, further insight into the biological toxicity caused by PM was gleaned by investigating the expression of pro-inflammatory mediators and receptors. Specifically, it was determined that AhR was constantly degraded, resulting in the overexpression of CYP1A1 in HaCaT cells, upon treatment with PM. Indeed, it is well known that AhR undergoes degradation upon translocation from the nucleus to the cytoplasm after activation. Therefore, the reduction level of AhR in terms of the dose-/time-dependency on PM treatment was obviously apparent. The upregulating expression of pro-inflammatory proteins (MMP-9 and COX-2), as well as MAPK signal (ERK1/2) also was obvious. Overall, the results suggested that PM is a potential risk to human health, especially the skin.

Fortunately, the effective antioxidant properties of MA-OHE have been successfully applied to protect the skin against harmful PM. The present results revealed that MA-OHE inhibited the degradation of AhR, downregulated the expression of pro-inflammatory proteins, and reduced MAPK signals caused by PM exposure. A possible explanation is that MA-OHE was absorbed by HaCaT keratinocytes and that it subsequently downregulated the phosphorylating transcription factors (ERK, JNK, and p53) and scavenged ROS radicals, thereby repairing damaged DNA and protecting skin cells from environmental factors. In addition, comparing data between MA-OHE and raw *O. humifusa* demonstrated that MA-OHE had a lower viscosity and higher antioxidant activity than its precursor. This was because the antioxidant compounds in the raw *O. humifusa* may be released due to viscosity reduction by microwave treatment. Furthermore, the data about the reducing efficiency of MA-OHE on PM-induced genotoxicity and ROS generation indicated that MA-OHE had a better performance than raw *O. humifusa.* Therefore, it could be concluded that the microwave-assisted extraction method upgraded the bioactivities of raw *O. humifusa* in the resultant MA-OHE, and with its advanced physical properties, high biocompatibility, excellent antioxidant activity, and anti-pollution ability, MA-OHE could be a promising candidate for cosmetic applications, especially for PM protection products.

## 5. Conclusions

The results of this study showed that MA-OHE was able to overcome the physicochemical disadvantages of raw *O. humifusa* effectively thanks to its enhanced antioxidant properties and reduced viscosity. It was also found that MA-OHE could suppress AhR degradation, ROS production, and COX-2 and MMP-9 expression through downregulation of the MAPKs-mediated (ERK1/2) signaling pathways in HaCaT keratinocytes due to PM exposure, a finding that was consistent with its antioxidant and anti-inflammatory activities. In conclusion, it was suggested that MA-OHE could potentially be used as a cosmeceutical ingredient to prevent PM-induced skin oxidative stress and inflammation.

## Figures and Tables

**Figure 1 antioxidants-09-00271-f001:**
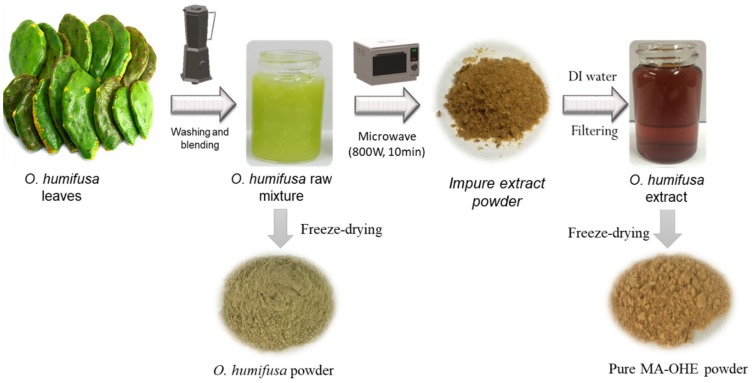
Schematic illustration of the microwave-assisted *O. humifusa* extract (MA-OHE) process.

**Figure 2 antioxidants-09-00271-f002:**
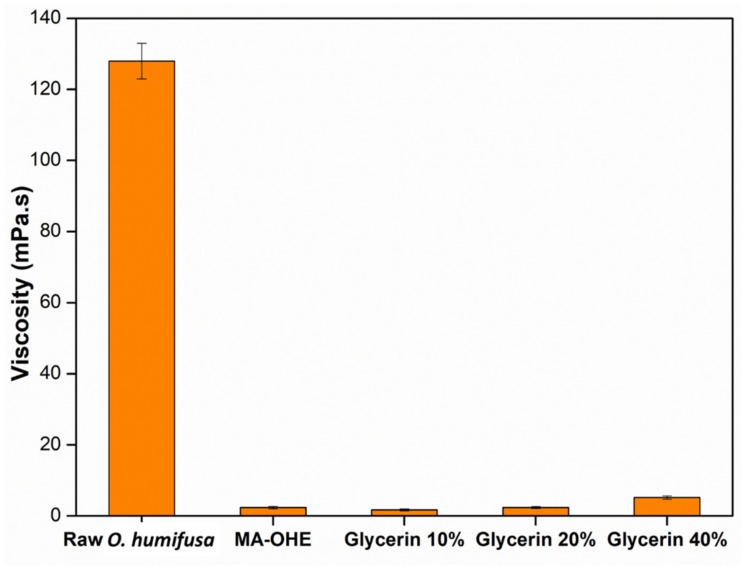
Viscosity values of raw *O. humifusa*, MA-OHE, and glycerin (10%, 20%, and 40%).

**Figure 3 antioxidants-09-00271-f003:**
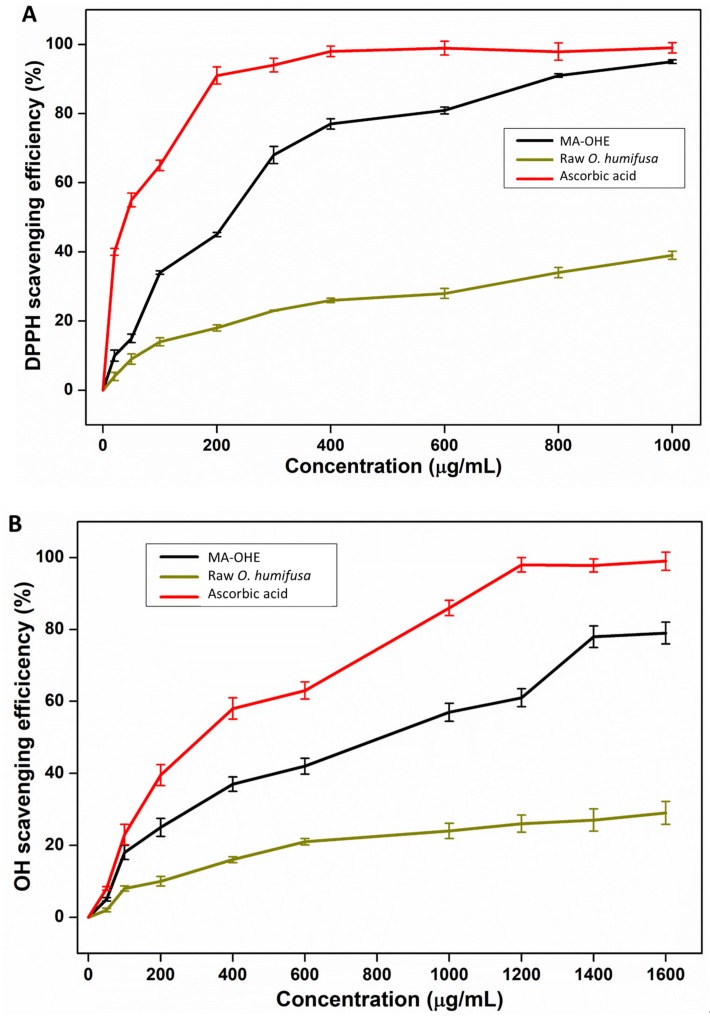
(**A**) DPPH radical scavenging activity and (**B**) OH radical scavenging activity of MA-OHE and raw *O. humifusa*.

**Figure 4 antioxidants-09-00271-f004:**
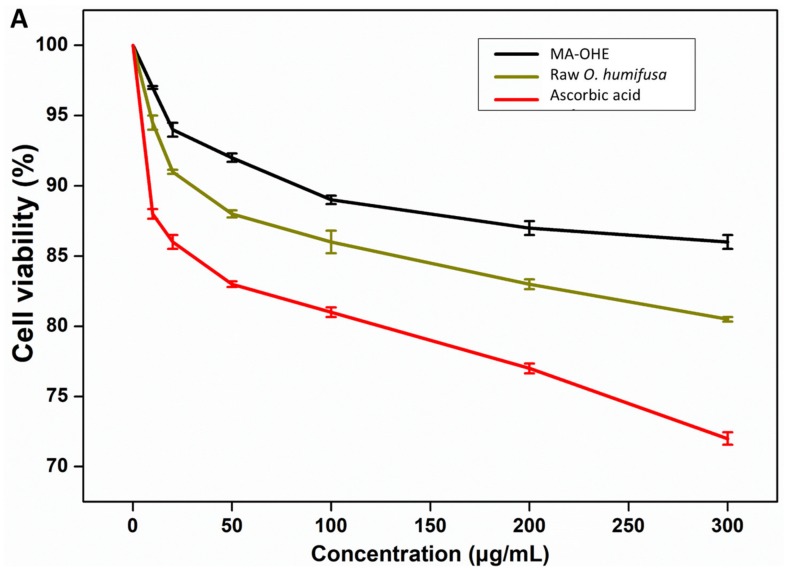
(**A**) Cell viability (%) of HaCaT cells in terms of MA-OHE, raw *O. humifusa*, and PM treatment for 24 h; (**B**) the effect of MA-OHE and raw *O. humifusa* in reducing the toxicity of PM on HaCaT cells. The error bars represent the standard deviation (SD) of triplicate measurements.

**Figure 5 antioxidants-09-00271-f005:**
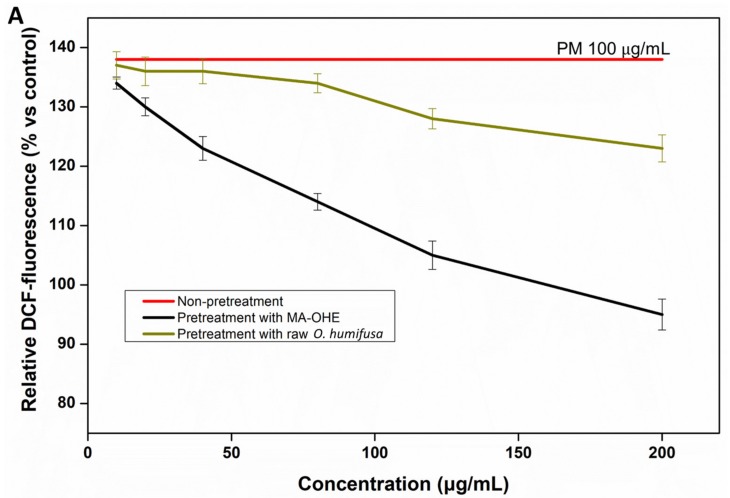
(**A**) The effects of MA-OHE and raw *O. humifusa* on the reduction of PM-induced ROS and (**B**) apoptosis incidence caused by PM exposure, MA-OHE and raw *O. humifusa* pretreatment on HaCaT cells after 24 h treatment.

**Figure 6 antioxidants-09-00271-f006:**
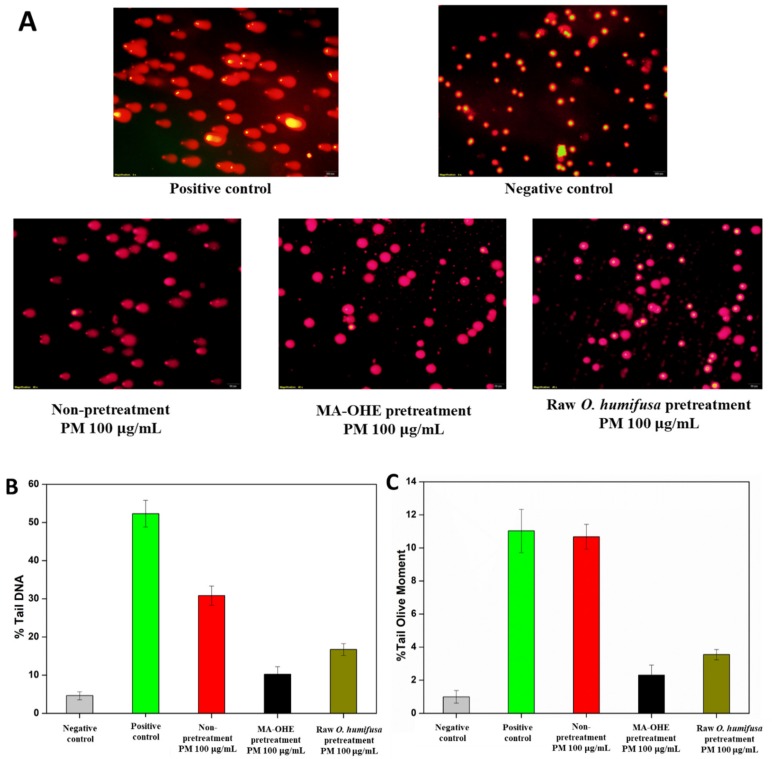
Evaluation of the genotoxicity on HaCaT cells caused by MA-OHE and PM after 24 h of treatment by using the comet assay: (**A**) fluorescence images representative of HaCaT cells after electrophoresis (Scale bar = 250 µm); (**B**) tail DNA damage (%); (**C**) tail olive moment (%) (*p* < 0.05, significant difference is in comparison with the control).

**Figure 7 antioxidants-09-00271-f007:**
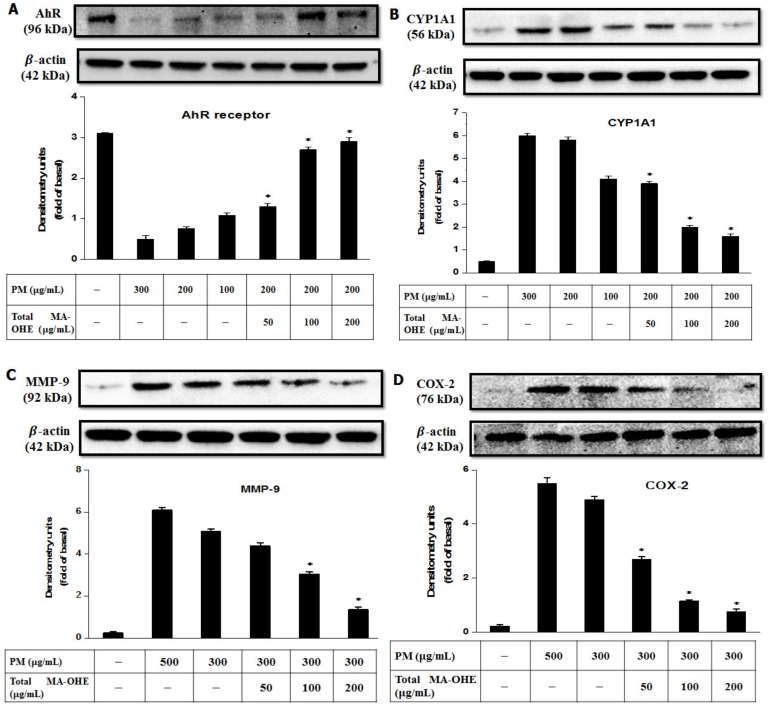
Expression of AhR receptors: (**A**) AhR and (**B**) CYP1A1); inflammatory proteins (**C**) COX-2, (**D**) aging protein (MMP-9), and (**E**) MAPK signaling protein (ERK1/2) following various MA-OHE pretreatments in PM-exposed HaCaT keratinocytes (* *p* < 0.05, significant difference from PM-induced-only group; experiments were performed in triplicate).

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
