# Peer review of "Effects of Microwave-Assisted Opuntia humifusa Extract in Inhibiting the Impacts of Particulate Matter on Human Keratinocyte Skin Cell"

_antioxidants, 2020, doi:10.3390/antiox9040271_

Round 1
Reviewer 1 Report
Thank you very much for submitting your manuscript ID No.antioxidants-740284 “ Article Title: Effects of Microwave-Assisted Opuntia Humifusa Extraction in Inhibiting Impacts of Particulate Matter on Human Skin” for Antioxidants. I think this manuscript is provided us with interesting information on a new extract. Unfortunately, this manuscript is a lack of standardized analytical data excluding viscosity, because of producing a stable extract. I hope to add the information.
I will propose the other suggestions to the authors.
- Title: “Human keratinocyte skin cell” is correct, because this is based on in vitro study using HaCaT cells.
- Fig.4(A): The cell viability of only PM should be addressed. Fig.4 (B): It is not useful information and is deleted.
- Subtitle: “3.4 Effect of MA-OHE on reducing the toxicity of PM on human skin to keratinocyte skin cells” is correct.
- Fig.9: This manuscript is not a review paper. Fig.9 is deleted including no data information in this manuscript.
According to my suggestions, I wish the authors will revise your one soon. Thank you.
Author Response
Reviewer 1
- This manuscript is a lack of standardized analytical data excluding viscosity, because of producing a stable extract. I hope to add the information.
Thank you for your important comment. After evaluating the data, we made a change in the structure of the manuscript and added some crucial information. I hope that the current version gives you a complete satisfaction. Please take a look at the full manuscript again.
- Title: “Human keratinocyte skin cell” is correct, because this is based on in vitro study using HaCaT cells.
Thank you so much for your suggestion. We corrected the title
- New title: Effects of Microwave-Assisted Opuntia Humifusa Extract in Inhibiting Impacts of Particulate Matter on Human Keratinocyte Skin Cell
- Fig.4(A): The cell viability of only PM should be addressed. Fig.4 (B): It is not useful information and is deleted.
Thank you so much for your comment. We added the cell viability of only PM. We also changed Figure 4A and 4B
- Subtitle: “3.4 Effect of MA-OHE on reducing the toxicity of PM on human skin to keratinocyte skin cells” is correct.
Thank you so much for your suggestion. We corrected it
- Fig.9: This manuscript is not a review paper. Fig.9 is deleted including no data information in this manuscript.
Thank you so much for your suggestion. We removed Fig. 9
Reviewer 2 Report
The manuscript addresses a topical and relevant subject and is undoubtedly interesting, but several aspects should be improved before publication.
- There is a general confusion regarding the abbreviation MA-OHE, which in some points refers to the extraction and in others to the extract. I would suggest authors to use this abbreviation only for the extract. Also in the title, I would replace “Extraction” with “Extract”.
- Some important experimental details are missing:
- How was the raw humifosa extract prepared?
- What was the yield of MA-OHE and how were the solutions at defined concentrations prepared for the antioxidant and cellular assays? Similar details should be provided also for the raw extract in order to understand how to compare the results obtained for the two samples in the different assays.
- It is not clear how PM were prepared and administered to the cells.
- The experimental protocol used for ROS detection (Figure 5A) seems to be not reported in Materials and Methods.
- Table S1 is of not use: it is not rigorous, quantitative data concerning the antioxidant properties are missing (the concentrations of the extracts needed to achieve the reported antioxidant activities should be added), and the terms “High” and “Low” can not be accepted for a scientific comparison. The only useful data are those regarding the yields.
- I was not able to catch the significance of paragraph 3.3.: it is almost like authors are making a comparison in terms of toxicity between MA-OHE and PM. Instead, the effects of MA-OHE against the damages induced by PM should be reported together with data from control samples as in paragraph 3.4. Only Figure 4A should be left as such.
- Several comments in the Discussion section seem gratuitous and not supported by experimental data, see for example page 14, lines 5-6; page 14, lines 8-10; page 14, lines 17-23.
- The antioxidant properties of raw Humifosa extract should be tested as well to state that MA-OHE exhibits enhanced antioxidant properties.
- Editing of English language and style is required in some points because some parts of the manuscript (e.g. abstract) are difficult to understand and this lowers the quality of the manuscript. (see also page 1, lines 34-35; use of the term “ antioxidation”; page 7, lines 12-13; Figure 9)
Some other minor points:
- Page 7, lines 8-9: this seems true only for the DPPH assay.
- Page 10, lines 10-13: do authors refer to data reported in Figure 4?
- Figure 7: authors should have used higher concentrations of PM in order to induce a more extensive damage and hence demonstrate more significant protective effects of MA-OHE.
- Page 11, lines 17-20: actually, differences between 2 h and 3 h pretreatments seem not be negligible for most MA-OHE doses.
- Page 14, line 39: which types of quinones are present in PM?
- Page 15, lines 28-29: actually, data on the suppression of PM-induced ROS production by MA-OHE seem not to be reported in the manuscript.
Author Response
Reviewer 2
- There is a general confusion regarding the abbreviation MA-OHE, which in some points refers to the extraction and in others to the extract. I would suggest authors to use this abbreviation only for the extract. Also in the title, I would replace “Extraction” with “Extract”.
Thank you so much for your suggestion.
We corrected the title and used “extract” expression for the abbreviation MA-OHE throughout the manuscript.
- New tile: “Effects of Microwave-Assisted Opuntia Humifusa Extract in Inhibiting Impacts of Particulate Matter on Human Keratinocyte Skin Cell”
- Some important experimental details are missing:
Thank you so much for your comments. We revised and supplied the needed information
- How was the raw O. humifosa extract prepared?
For preparation of raw O. humifusa, fresh and healthy leaves of O. humifusa were thoroughly washed with water, air dried, and chopped into small pieces. The chopped leaves (20 g) were then ground into a homogenous mixture in a blender. The resulting product has a very high viscosity, which is directly used for viscosity measurement and production of MA-OHE extract. For further experiments, it is diluted with DI water in various concentrations.
- What was the yield of MA-OHE
The extraction yield Y (%) of MA-OHE was calculated as follows:
(1)
Where, W1 is the weight (g) of pure MA-OHE powder, while W2 is the weight (g) of initial O. humifusa powder. The calculated result indicated that the extraction yield is about 33.5%.
- How were the solutions at defined concentrations prepared for the antioxidant and cellular assays? Similar details should be provided also for the raw extract in order to understand how to compare the results obtained for the two samples in the different assays.
Thank you so much for your comments. We added the needed information
Because the extraction yield is about 33.5%, the raw O. humifusa’s concentration triples that of the pure MA-OHE powder in the antioxidant activity and cellular assays.
2.5.1. Preparation of samples
The antioxidant activity of the pure MA-OHE powder was tested by DPPH-radical and hydroxyl radical assays. The pure MA-OHE powder was dissolved and diluted in DI water to specific concentrations under gentle shaking. In comparison, the raw O. humifusa samples were simultaneously tested antioxidant activity assays. The raw O. humifusa was also dissolved in DI water with a high-speed stirring. However, corresponding to each concentration of the pure MA-OHE powder, the raw O. humifusa’s concentration was tripled. This preparation method was also used for cell viability assay.
- It is not clear how PM were prepared and administered to the cells.
Thank you for your comment. We added the information
According to our previous study [3], PM samples were collected from subway stations in Seoul, Korea. The samples were collected on Φ 47 mm quartz filters (QMA filter, Φ 47 mm, Pall Corporation, NY, USA) utilizing a Ze- flour filters (polytetrafluoroethylene membrane filter, Φ 47 mm, Pall Corp.) and a mini-volume air sampler (TAS, 5 litter per min, Airmetrics, Eugene, OR, USA) using a low-volume air sampler (PMS-104, 16.7 litter per min, APM, Bucheon, Korea). PM samples were dispersed in DI water for 30 min by using a sonicate bath and the PM dispersions were used within 1 h to avoid the degradation of PM components.
PM samples were dispersed in DI water for 30 min by using a sonicate bath and the PM dispersions were used within 1 h to avoid the degradation of PM components. Then, the initial PM dispersions were diluted to final concentrations (10-200 g/mL) by using DMEM medium.
- The experimental protocol used for ROS detection (Figure 5A) seems to be not reported in Materials and Methods.
2.8. Evaluation of intracellular reactive oxygen species (ROS)
HaCaT cells (2.5 × 105cells/mL) seeded in a black 96-well-plate (Thermo Scientific™, USA) were treated by MA-OHE, raw O. humifusa and PM at concentration of 0-200μg/mL along with 10μM of dipyridamole-a radical scavenger as a negative control under identical conditions [3]. After 1 and 2 days of incubation, 10μM DCFH-DA dissolved in DMSO was stained for 1 h in a darkroom, and then twice washed with PBS buffer. In the literature, DCFH-DA penetrates into the internal cells and then is hydrolyzed into DCFH by esters. Immediately, this was recorded by fluorescence measurement at 485/530 nm.
- Table S1 is of not use: it is not rigorous, quantitative data concerning the antioxidant properties are missing (the concentrations of the extracts needed to achieve the reported antioxidant activities should be added), and the terms “High” and “Low” cannot be accepted for a scientific comparison. The only useful data are those regarding the yields.
Thank you for your comments. After evaluating the data, we removed Table S1
- I was not able to catch the significance of paragraph 3.3.: it is almost like authors are making a comparison in terms of toxicity between MA-OHE and PM. Instead, the effects of MA-OHE against the damages induced by PM should be reported together with data from control samples as in paragraph 3.4. Only Figure 4A should be left as such.
Thank you so much for your very important comments. We integrated paragraph 3.3 and 3.4 and concentrated on discussing the effects of MA-OHE and raw O. Humifosa against the PM-induced damages. We also added some crucial data of raw O. Humifosa as a control sample. Please take a look at the paragraph 3.3 again.
- Several comments in the Discussion section seem gratuitous and not supported by experimental data, see for example page 14, lines 5-6; page 14, lines 8-10; page 14, lines 17-23.
Thank you for your comments. We corrected and supplied some information and references for supporting the discussions
- Page 14, lines 5-6: According to the results, the MA-OHE presented not only low viscosity but also enhanced antioxidant capacity relative to raw O. humifusa.
We tested the antioxidant activity of raw O. humifusa to support for this conclusion. Please see the Figure 3 in the manuscript.
- Page 14, line 8-10: As compared with the conventional methods, MA-OHE exhibited several remarkable advantages, including: (i) significantly higher yield; (ii) lower viscosity; (iii) considerably higher antioxidant activity.
We corrected and added the references to support for this.
As compared with the conventional methods, MA-OHE exhibited several remarkable advantages regarding to yield, viscosity, and antioxidant activity [11,12,17,21]. For illustration, the MA-OHE showed a significantly higher yield (33.5%), compared with extracts prepared by conventional methods such as solvent extraction and hot water method (10.5 – 23.1%) [11,17]. The data also indicated that MA-OHE possesses a higher antioxidant activity, nearly 98% of DPPH radical scavenging efficiency, while the maximum percentage of DPPH radical scavenging efficiency reported in other studies was only 80% [31]. While MA-OHE showed similar effects to those of other advanced technologies (e.g., ultrasound-assisted extraction-UAE) in terms of yield (UAE: 31.7%) [12]. However, MA-OHE possessed a lower viscosity as compared with the UAE method. This was due to the fact that the acoustic energy (a mechanical energy) used in the UAE process is not absorbed by molecules, but is only transmitted throughout the medium [21], whereas the microwave method uses high energy to heat molecules by a dual mechanism of ionic conduction and dipole rotation, which reduces the attractive forces between the molecules.
- Page 14, line 17-23: A possible explanation is that under high-temperature and high-energy treatment, the O. humifusa was transformed into carbon dots (CDs) and enhanced its antioxidant amount [22]. Additionally, the very high viscosity can prevent the release of antioxidants in raw O. humifusa. In addition to the initial antioxidant components (flavonoids and phenolic compounds), the synthesized CDs also are potential bioactive compounds that can improve the antioxidant properties of MA-OHE. Overall, microwave-assisted extraction helps to maintain nutrition, improve texture and appearance, and reduce viscosity relative to the conventional extraction methods.
We corrected and adjusted it
A possible explanation is that under high-temperature and high-energy treatment, the raw O. humifusa was transformed into carbon dots (CDs) [22] and significantly reduced the vicosity. It is supposed that it is the viscosity reduction that allow MA-OHE can release antioxidant compounds, which resulted in a higher antioxidant activity in MA-OHE, compared with raw O. humifusa. Moreover, it was demonstrated that the plant extract-based CDs prepared by the microwave assisted process have low toxicity [22]. Overall, microwave-assisted extraction method helps to maintain nutrition, improve texture and appearance, and reduce viscosity relative to the conventional extraction methods.
- The antioxidant properties of raw Humifosaextract should be tested as well to state that MA-OHE exhibits enhanced antioxidant properties.
Thank you for your comment. We added the antioxidant assay for raw O. Humifosa
- Editing of English language and style is required in some points because some parts of the manuscript (e.g. abstract) are difficult to understand and this lowers the quality of the manuscript. (see also page 1, lines 34-35; use of the term “antioxidation”; page 7, lines 12-13; Figure 9)
Thank you for your comments. We corrected all of them
- Page 1, line 34-35: It is primarily classified based on its diameter, which can be visualized in terms of the sizes of human hair and fine beach sand ( 70 and 90 m, respectively): PM10 ( 10 m) and PM2.5 (2.5 m)
We corrected it
Based on the diameter that can cause harmful effects to human, PM primarily classified into two group: PM10 ( 10 m) and PM2.5 (2.5 m)
- Use of the term “antioxidation”
We replaced “antioxidantion” by “antioxidant activity”
- Page 7, lines 12-13: In the present study, the MA-OHE extract also showed lower EC50 values for the DPPH· and OH· radical assays at 300 and 600 µg/ml, respectively, compared with ascorbic acid.
We corrected it
The MA-OHE reached EC50 values for both DPPH· and OH· radical assays at 300 and 600 µg/ml.
- Figure 9: We removed Figure 9
- Some other minor points:
Page 7, lines 8-9: this seems true only for the DPPH assay.
Generally, the inhibitory activities of MA-OHE extract are rapidly increased at concentrations below 400 µg/ml, after which the reaction slows down significantly until the highest concentration
Thank you for your comments
After checking the Figure 3, this statement is for both DPPH and OH radical assays
Page 10, lines 10-13: do authors refer to data reported in Figure 4?
Figure 7: authors should have used higher concentrations of PM in order to induce a more extensive damage and hence demonstrate more significant protective effects of MA-OHE.
Thank you for your comments.
We changed the Figure 7 by integrated with Figure 4 and added a comparison between MA-OHE and raw O. humifusa in reducing the PM cytotoxicity. Please see the Figure 4B in the current manuscript.
Page 11, lines 17-20: actually, differences between 2 h and 3 h pretreatments seem not be negligible for most MA-OHE doses.
Thank you for your comment.
After evaluating the data, we only used pretreatment condition for 2h.
Page 14, line 39: which types of quinones are present in PM?
The four quinones including: 1,2- naphthoquinone (1,2-NQ), 1,4-naphthoquinone (1,4-NQ), 9,10-phenanthraquinone (9,10-PQ), and 9,10-anthraquinone (9,10-AQ) was demonstrated to be present in particulate matter
You can see more detail in the reference: “Cho et al. Determination of Four Quinones in Diesel Exhaust Particles, SRM 1649a, and Atmospheric PM2.5. Aerosol Science and Technology, 38(S1):68–81, 2004”
Page 15, lines 28-29: actually, data on the suppression of PM-induced ROS production by MA-OHE seem not to be reported in the manuscript.
Thank you so much for your comment
We added the data, please see the Figure 5A in the current manuscript.

Round 2
Reviewer 2 Report
The authors have satisfactorily addressed my comments, therefore I can now reccomend pubblication of the manuscript.
However, I still have a concern regarding the use of "dipyridamole-a radical scavenger as a negative control under identical conditions" in the evaluation of intracellular reactive oxygen species. It is not clear to me what authors want to prove by using this compound, also because data on its effect seem not to be reported in the manuscript.
Author Response
Dear reviewer
We sincerely appreciate your valuable comments, which allow us make the manuscript more perfect for the readers of Antioxidants.
Your comment: However, I still have a concern regarding the use of "dipyridamole-a radical scavenger as a negative control under identical conditions" in the evaluation of intracellular reactive oxygen species. It is not clear to me what authors want to prove by using this compound, also because data on its effect seem not to be reported in the manuscript.
Thank you so much for your comments. We revised and supplied the needed information
2.8. Evaluation of intracellular reactive oxygen species (ROS)
The generation of intracellular ROS was measured by DCFH-DA assay. Briefly, HaCaT cells (2.5 × 105cells/mL) seeded in a black 96-well-plate (Thermo Scientific™, USA) were pretreated by MA-OHE and raw O. humifusa at concentration of 0-200μg/mL for 2h; then exposed with PM (100 μg/mL) [3]. After 1 and 2 days of incubation, 10μM DCFH-DA dissolved in DMSO was stained for 1 h in a darkroom, and then twice washed with PBS buffer. In the literature, DCFH-DA penetrates into the internal cells and then is hydrolyzed into DCFH by esters. Immediately, this was recorded by fluorescence measurement at 485/530 nm. Dipyridamole was used as a control with the relative ROS level of 100%. The ROS generation was calculated as formula (4)
(4)
Where F is fluorescence intensity of cells pretreated with MA-OHE and raw O. humifusa; Fo is fluorescence intensity of dipyridamole
